# Exploring Charged Polymeric Cyclodextrins for Biomedical Applications

**DOI:** 10.3390/molecules26061724

**Published:** 2021-03-19

**Authors:** Noemi Bognanni, Francesco Bellia, Maurizio Viale, Nadia Bertola, Graziella Vecchio

**Affiliations:** 1Dipartimento di Scienze Chimiche, Università degli Studi di Catania, Viale A. Doria 6, 95125 Catania, Italy; noemibognanni91@gmail.com; 2Istituto di Cristallografia, CNR, via P. Gaifami 18, 95126 Catania, Italy; francesco.bellia@cnr.it; 3IRCCS Ospedale Policlinico San Martino, U.O.C. Bioterapie, L.go R. Benzi 10, 16132 Genova, Italy; maurizio.viale@hsanmartino.it (M.V.); nadia.bertola@gmail.com (N.B.); 4Consorzio Interuniversitario di Ricerca in Chimica dei Metalli nei Sistemi Biologici (CIRCMSB), via Celso Ulpiani, 27, 70126 Bari, Italy

**Keywords:** aggregation, cancer, doxorubicin, nanoparticles

## Abstract

Over the years, cyclodextrin uses have been widely reviewed and their proprieties provide a very attractive approach in different biomedical applications. Cyclodextrins, due to their characteristics, are used to transport drugs and have also been studied as molecular chaperones with potential application in protein misfolding diseases. In this study, we designed cyclodextrin polymers containing different contents of β- or γ-cyclodextrin, and a different number of guanidinium positive charges. This allowed exploration of the influence of the charge in delivering a drug and the effect in the protein anti-aggregant ability. The polymers inhibit Amiloid β peptide aggregation; such an ability is modulated by both the type of CyD cavity and the number of charges. We also explored the effect of the new polymers as drug carriers. We tested the Doxorubicin toxicity in different cell lines, A2780, A549, MDA-MB-231 in the presence of the polymers. Data show that the polymers based on γ-cyclodextrin modified the cytotoxicity of doxorubicin in the A2780 cell line.

## 1. Introduction

Cyclodextrins (CyDs) are cyclic oligosaccharides of α-1,4-linked D(+)-glucopyranose with the unique property to act as molecular containers. They have been used in biomedical applications for their ability to include drugs or several biomolecules, such as cholesterol [1,2,3,4].

CyD properties can be modulated through their chemical modification. In recent years, CyDs represented an important nanocarrier family, thus developing into sophisticated drug delivery systems [5,6,7]. Nanoparticles based on CyD, have allowed encapsulation of drugs to protect them and improve their bioavailability. These systems have higher stability constants than those of the corresponding CyD monomers; they show excellent properties in drug release kinetics, mechanical properties and stimuli-responsiveness [8,9,10,11]. Furthermore, these polymeric systems can be used in a clinical setting, such as controlled drug and gene delivery systems [12]. Successful examples of linear CyD polymers specifically designed as drug carriers are Cyclosert and CALAA-01 [13]. CyD polymers modified with choline, amino or carboxylic groups have been investigated to increase their drug loading ability.

Due to the ability to include aromatic molecules of appropriate sizes, CyDs have also been investigated as protein chaperones [14,15,16]. β-CyD reduced the β-amyloid aggregation in vitro at millimolar concentration. The protective effects of β-CyD were also proven in vivo. The interaction between Amyloid beta peptide (Aβ) and α-, β- and γ-CyD was correlated to the ability to include Phe19 and Phe20 side chain of Aβ [16]. Other studies have suggested the higher antiaggregant activity of some functionalised CyDs with aromatic moieties [17]. CyD dimers were also found to act as inhibitors of Aβ_40_ aggregation at 1 mM concentration [18,19]. We investigated amino-cyclodextrin oligomers as promising CyD derivatives in inhibiting the aggregation of Aβ at micromolar concentration [20].

Inspired by the properties of multi-cavity systems and the importance of functionalisation to improve CyD properties, in this paper we report the synthesis of new linear polymers of β- and γ-CyD with different contents of guanidinium positive charge and number of CyD cavities. We also assayed the polymer systems as antiaggregant agents and as drug carriers to explore the effects of multi-cavity systems and their functionalisation for biological applications (Figure 1). 

It is well known that cancer cells have a deficiency of Arg amino acid and their requirement for arginine is higher than that for other amino acids. Certain tumours lose the ability to synthesise arginine dependently. Therefore, arginine depletion can be considered the weak point in cancer treatment for arginine auxotrophic tumours [21,22]. We used doxorubicin (DOX) as a drug model to test the activity of the new CyD derivatives as nanocarriers. Several studies have highlighted that CyDs can stabilise DOX in solution and enhance its dissolution rate [23,24]. Complexation of DOX with CyDs can increase permeability across the blood–brain barrier, due to the disruption of the membrane [25].

## 2. Results and Discussion

### 2.1. Synthesis and Characterisation 

Charged CyD polymers were synthesised in water, starting from the polypeptide N-butyl-polyglutamate (PGA), ArgOCH_3_ and CyD 3-amino derivatives, using 4-(4,6-Dimethoxy-1,3,5-triazin 2-yl)-4-methylmorpholinium chloride (DMTMM) as the condensing agent, as reported elsewhere [26]. We found that this method gave high conjugation yield using a green synthetic route. This procedure is appropriate for modulating the number of CyD cavities and charged groups of Arg in the PGA backbond [27]. 

We synthesised various polymers with different amounts of Arg and CyD units to explore the effect on polymer properties.

All the new polymers were characterised by NMR (Appendix A). In Figure 2 the NMR spectra of PGAβCyDArg1 and PGAβCyDArg4 are reported. ^1^H NMR spectra of all the derivatives show common patterns; the protons of CyD resonate at 5 ppm (H-1), and 4.0-3.4 ppm (H-3,-6,-5,-4,-2). Protons of arginine and the glutamic acid side chain of PGA resonate at 3.3 ppm and 2.5-1.8 ppm region. Butyl protons of PGA are also evident between 1.5 ppm and 1.0 ppm. We determined the number of CyD units linked to the PGA backbone for each polymer derivative by calculating the integral ratios of the signal of Hs-1 of CyD, the signal of the ethylene chain protons of PGA or the N-buthyl chain protons. Moreover, the integral ratio of signal due to the γ-CH_2_ of Arg moieties at 3.3 ppm and the signals of PGA ethylenic protons or the N-buthyl protons was used to value the number of Arg moieties grafted to the polymer. The results obtained from NMR for each bioconjugate are reported in Table 1.

The ^13^C NMR spectra of the derivatives show signals due to guanidium carbons at about 160 ppm and signals around 174 ppm due to the carboxyl group of PGA and arginine methyl ester, in addition to the signals of CyD units in the aliphatic region. 

CyD polymers were also characterised by dynamic light scattering (DLS) and Zeta potential values were also measured (Table 1). The hydrodynamic diameters increase with the number of cavities linked to the PGA backbone. The Z potential values increased when the number of Arg units increased from negative values (−58 mV for PGA alone) to positive values, in keeping with the progressive increase in positive charges due to the guanidinium groups.

Spectrometric measurements were also carried out further to characterise the structural features of the new polymers. The MALDI spectra recorded in linear mode (Appendix A) mainly contain a wideband; the *m*/*z* values of the highest peaks match to those obtained by the NMR studies (Table 1), within the experimental errors, thus confirming the calculated molecular weight (Mw) of the new CyD polymers.

As for PGAβCyDArg1, the MALDI spectrum is resolved into several components (Figure 3). The average difference between two successive relative peaks is 1280 ± 20. This value suggests that the repeat unit contains both the CyD (MW 1135) and glutamic acid (MW 147) moieties, as expected. 

### 2.2. Antiaggregant Activity

The abnormal aggregation of Aβ is one of the main hallmarks of Alzheimer’s disease (AD). In the pathological pathway, the amyloid peptide firstly forms soluble and highly toxic oligomers; then, the growing dimensions of the aggregated species lead to the formation of fibrillary and insoluble structures, mainly accumulated into brain plaques [28]. Finding new molecules able to inhibit the formation of amyloid-type aggregates represents an important strategy to prevent the onset of AD or attenuate the development of this devastating disorder. Therefore, we tested the effect of all the PGA polymers on the self-induced aggregation of Aβ by using a turn-on fluorescent dye Thioflavin T (ThT), sensitive to the formation of fibril species.

The fluorescence data recorded for the amyloid-type aggregation of Aβ fit to a sigmoid curve. The maximum fluorescence gain (*F_max_* − *F*_0_) is 33 ± 1 and the lag phase lasts 24 ± 2 h. When the compounds of interest are co-incubated with Aβ, the kinetic parameters of the aggregation process could be modified due to non-covalent interactions between Aβ and the PGA polymers. *F_max_* − *F*_0_ is proportional to the amount of Aβ fibrils, whereas during the lag phase (*t_lag_*) only small aggregated species form. As a consequence, the lower *F_max_* − *F*_0_ is and/or the higher *t_lag_* is, the better the antiaggregant activity.

Figure 4 shows the *F_max_* − *F*_0_ values obtained by the aggregation of Aβ alone (control (CTRL)) or in the presence of each PGA polymer. Several amounts of the compounds have been tested, the (Aβ)/(Polymer) molar ratio ranging from 1:1 to 1:8. 

The co-incubation of any polymer with Aβ in a 1:1 molar ratio induced little or no effect on the final extent of the aggregation process. Higher amounts of the polymers significantly decrease the *F_max_* − *F*_0_ values and the antiaggregant activity is exerted in a dose-dependent manner, as reported in Figure 5 in the case of PGAβCyDArg2.

As for the PGA polymers containing β-CyD (PGAβCyDArg1, PGAβCyDArg2 and PGAβCyDArg4), the antiaggregant activity is very comparable among the polymers, when the Aβ/polymer molar ratio was 1:2. Instead, these β-CyD-containing polymers differently affect the amyloid aggregation extent when the 1:8 Aβ/polymer molar ratio was tested. In particular, it seems that the greater the number of Arg units, the greater the inhibition effect of the amyloid aggregation. Such a trend is also observed when the γ-CyD-containing polymers were taken into account (PGAγCyDArg3 and PGAγCyDArg5). However, the inhibition activity of these γ-CyD polymers is slightly lower than that shown by the corresponding β-CyD polymers (PGAβCyDArg2 and PGAβCyDArg4). Such a difference could be reasonably ascribed to the structural differences between the β- and γ-CyD cavities that, in turn, could affect the non-covalent interaction between the amyloid peptide and the polymers. A similar trend has been reported for single CyDs [14].

The lag phase of the amyloid aggregation was not significantly modified by the polymers (data not shown), meaning that the interaction between the Aβ and these PGA polymers is not altered by the number of CyD and Arg units.

Above all, the antiaggregant activity towards the self-induced formation of amyloid aggregated species exerted by the PGACyD polymers could reasonably be due to the effects of both the CyD cavity and charged units (Arg and PGA). The effect of positive and negative charges of carbohydrate polymers on the on-pathway Aβ aggregation has been recently ascertained [29], thus corroborating these results.

### 2.3. Solubility Experiments

We explored the different affinity of the polymers for the guest DOX by solubility experiments. The effect of the polymers on the solubility can provide a comparison of the affinity for a guest. Data were reported in Appendix A. We found that water solubility of DOX (2.2 × 10^−4^ M) increased in the presence of all the polymers at concentration 25 mg/mL. Particularly the polymers based on γ-CyD showed an effect higher than that of β-CyD polymers at physiological (pH 7.4). PGAγCyDArg4 with more γ-CyD cavities was also more effective than that with a lower numerv of CyDs. This trend is in keeping with the highest affinity of γCyD cavities for DOX [26,30].

### 2.4. Antiproliferative Activity (MTT Assay)

New polymers were studied as drug delivery systems for the topoisomerase inhibitor DOX. We performed cell proliferation assays on A2780, A549 and MDA-MB-231 cancer cell lines. Complexes polymer/DOX (1:10 molar ratio) were prepared and assayed. Data obtained are reported in Table 2. Polymers alone did not show toxicity for cells (data not shown). 

Overall, the data show that the polymers did not change the antiproliferative activity of DOX significantly. In fact, in A549 and MDA-MB-231 cell lines, the half maximal inhibitory concentration (IC_50_) values do not change significantly depending on the type of functionalisation. Conversaly, in A2780 cells the complexes with PGAγCyDArg3 (*p* = 0.0028), and PGAβCyDArg5 (trend, *p* = 0.0738) showed higher IC_50_ values compared to free DOX. 

In the case of the polymers based on γ-CyD, the higher affinity for DOX suggested by solubility data may explain the effect on the cytotoxicity, as reported for similar systems studied by us [26]. The reduction in the antiproliferative effect was observed for many systems and only in vivo studies may provide information on the potential of the drug carriers [31].

Only PGAβCyDArg1/DOX showed a slight trend towards a higher antiproliferative activity (*p* = 0.0657) than free DOX and a significant difference compared to PGAγCyDArg3, PGAγCyDArg4, and PGAβCyDArg5 (Table 2). 

## 3. Materials and Methods

### 3.1. Materials 

The water-soluble polymer butyl-polyglutamate (20) sodium salt (3 KDa, PGA) was acquired from IRIS Biotech gmbh. 3A-amino-3A-deoxy-2A(S),3A(R)-β-cyclodextrin (βCyD3NH2), 3A-amino-3A-deoxy-2A(S),3A(R)-γ-cyclodextrin (γCyD3NH2) and DOX were acquired from TCI (Tokyo Chemical Industry). L-Arginine methyl ester dihydrochloride was acquired by SIGMA Aldrich. Aβ_1-40_ (Bachem) was properly treated, as previously reported [32]. Inclusion complexes of DOX with CyD polymers were prepared by mixing the stock solution of DOX with solutions of CyD polymers for 2 h.

#### 3.1.1. Synthesis of PGAβCyDArg1

βCyD3NH_2_ (50 mg in 1 mL of H_2_O), and DMTMM (18.26 mg in 350 μL) were added to PGA (6.61 mg in 350 μL) every 30 min in three aliquots. The pH of the reaction mixture was adjusted to 8. After 24 h, ArgOCH_3_(2 mg) and DMTMM (18 mg) were added to the solution (during 30 min). The reaction mixture was stirred at r.t. for 24 h. 

The polymer was isolated with Sephadex G-15 column chromatography. The various fractions collected were examined using TLC, (eluent PrOH/AcOEt/H_2_O/NH_3_ 5:2:3:1). The main product was characterised by NMR spectroscopy.

^1^H NMR (500, in D_2_O) δ (ppm): 5.20-4.80 (H-1 of CyD); 4.28 (s, CH Glu); 4.2 (m, CH Arg); 4.2-3.2 (m, H-3, -6, -5, -2,-4 of CyD, OCH_3_); 3.2 (m, γ CH_2_ Arg); 2.60–1.50 (m, β- and γ-CH_2_ PGA); 1.35 (m, CH_2_ butyl chain of PGA); 1.26 (m, CH_2_ butyl chain of PGA), 0.88 (m, CH_3_ butyl chain of PGA). 

^13^C NMR (125 MHz, in D_2_O) δ (ppm): 24.5 (β-CH_2_ of Arg), 26.8 (α-CH_2_ of Arg), 31.8 (β-CH_2_ of PGA), 40.5 (δ CH_2_ butyl chain of PGA), 52.3 (CH Arg), 52.9 (C-2 of CyD and OCH_3_ of Arg), 53.0 (CH of Glu), 60.0 (C-6 of CyD), 71.6 (C-3 of CyD ), 73.0 (C-5 of CyD), 80.4 (C-4 of CyD), 101-105 (C-1 of CyD), 160 (C=N of Arg), 173–174 (CNH PGA-CyD, PGA-Arg), 174.72 (CO methyl ester of Arg).

Dimension (DLS, Z average): 49 ± 5; PDI (DLS): 0.4; Zeta potential: 8 ± 1 mV (pH = 7.4).

The other polymers were synthesised in the same manner as PGAβCyDArg1 with different amounts of the reagents.

#### 3.1.2. Synthesis of PGAβCyDArg2

The synthesis was carried out as described above with PGA (7 mg), DMTMM (18 mg), ArgOCH_3_ (6 mg), DMTMM (10 mg) and βCyD3NH_2_ (50 mg).

^1^H NMR (500 MHz, in D_2_O) δ (ppm): 5.16-4.75 (H-1 of CyD); 4.28 (s, CH Glu); 4.20 (m, CH Arg); 4.10–3.20 (m, H-3, -6, -5, -2, -4 of CyD, OCH_3_); 3.13 (m, γ-CH_2_ Arg); 2.58-1.60 (m, γ-CH_2_ PGA,); 1.35 (m, CH_2_ butyl chain of PGA), 1.6 (m, CH_2_ butyl chain of PGA ); 1.28 (m, CH_2_ butyl chain of PGA), 0.89 (m, CH_3_ butyl chain of PGA).

Dimension (DLS, Z average): 35 ± 2 nm; PDI (DLS): 0.5; Zeta potential: 7.7 ± 0.5 mV (pH = 7.4).

#### 3.1.3. Synthesis of PGAγCyDArg3

The synthesis was carried out as for PGAβCyDArg2 with PGA (10 mg), DMTMM (19 mg), ArgOCH_3_ (8 mg), DMTMM (20 mg) and γCyDNH_2_ (67 mg).

^1^H NMR (500 MHz, in D_2_O) δ (ppm): 5.20–4.77 (m, H-1 of CyD); 4.32 (m, CH Arg); 4.23 (s, CH Glu); 4.12 (m, H-3A of CyD); 3.93–3.49 (m, H-3, -6, -5, -2, -4 of CyD and OCH_3_ of Arg); 3.12 (γ-CH_2_ Arg); 2.57–1.45 (β- and δ- CH_2_ PGA, CH_2_ Arg); 1.38 (m, CH_2_ butyl chain of PGA), 1.21 (m, CH_2_ butyl chain of PGA); 0.77 (m, CH_3_ butyl chain of PGA).

Dimension (DLS, Z average): 29 ± 3 nm; PDI (DLS): 0.4; Zeta potential: 2.3 ± 0.5 mV (pH = 7.4).

#### 3.1.4. Synthesis of PGAβCyDArg4 and PGAγCDArg5 

The synthesis was carried out as described above with PGA (25 mg), DMTMM (48 mg), ArgOCH_3_ (33 mg), DMTMM (21 mg) and βCyD3NH_2_ (59 mg) or γCyD3NH_2_ (72 mg).

##### PGAβCyDArg4

^1^H NMR (500 MHz, in D_2_O) δ (ppm): 5.22–4.80 (m, H-1 of CyD); 4.32 (m, CH Arg); 4.23 (s, CH Glu); 4.12 (m, H-3A of CyD); 3.93–3.50 (m, H-3, -6, -5, -2, -4 of CyD and OCH_3_ of Arg); 3.12 (γ-CH_2_ Arg); 2.60-1.49 (β- and δ- CH_2_ PGA, CH_2_ Arg); 1.39 (m, CH_2_ butyl chain of PGA), 1.20 (m, CH_2_ butyl chain of PGA); 0.78 (m, CH_3_ butyl chain of PGA).

^13^C NMR (125 MHz, in D_2_O) δ (ppm): 24.4 (β-CH_2_ of Arg), 27.6 (α-CH_2_ of Arg), 31.3 (β-CH_2_ of PGA), 40.5 (δ CH_2_ butyl chain of PGA), 52.34 (CH Arg), 52.8 (C-2 of CyD and OCH_3_ of Arg), 52.9 (CH of Glu), 60.0 (C-6 of CyD), 71.4 (C-3 of CyD), 72.5 (C-5 of CyD), 80.3 (C-4 of CyD), 101. 7 (C-1 of CyD), 160 (C=N of Arg), 173–174 (C-NH PGA-CyD, PGA-Arg), 174.7 (CO methyl ester of Arg). 

Dimension (DLS, Z average): 79 ± 8 nm; PDI (DLS): 0.6; Zeta potential: 45 ± 5 mV (pH = 7.4).

##### PGAγCyDArg5

^1^H NMR (500 MHz, in D_2_O) δ (ppm): 5.22–4.70 (m, H-1 of CyD); 4.32 (m, CH Arg); 4.23 (s, CH Glu); 4.12 (m, H-3-A of CyD); 4.05–3.32 (m, H-3, -6, -5, -2, -4 of CyD and OCH_3_ of Arg); 3.12 (γ-CH_2_ Arg); 2.59-1.46 (β- and δ- CH_2_ PGA, CH_2_ Arg); 1.37 (m, CH_2_ butyl chain of PGA), 1.21 (m, CH_2_ butyl chain of PGA); 0.78 (m, CH_3_ butyl chain of PGA).

^13^C NMR (125 MHz, in D_2_O) δ (ppm): 24.4 (β-CH_2_ of Arg), 27.7 (α-CH_2_ of Arg), 31.3 (β-CH_2_ of PGA), 40.4 (δ CH_2_ butyl chain of PGA), 52.4 (CH Arg), 52.8 (C-2 of CyD and OCH_3_ of Arg), 52.9 (CH of Glu), 60.0 (C-6 of CyD), 71.5 (C-3 of CyD), 72.0 (C-5 of CyD), 80.4 (C-4 of CyD), 101. 7 (C-1 of CyD), 160 (C=N of Arg), 173–174 (CNH PGA-CyD, PGA-Arg), 174.72 (CO methyl ester of Arg).

Dimension (DLS, Z average): 59 ± 6 nm; PDI (DLS): 0.6; Zeta potential: 37 ± 3 mV (pH = 7.4).

### 3.2. Instrumentation 

^1^H and ^13^C NMR spectra were recorded at 25 °C with a VARIAN UNITY PLUS-500 spectrometer at 499.9 and 125.7 MHz, respectively, using Varian library standard pulse programs. All samples were prepared in deuterated solvents (D_2_O); ^1^H NMR spectra were referred to the HOD signal and ^13^C NMR spectra to acetone (external reference). In all the experiments, the pulse at 90° lasted about 7 µs. 2D experiments (COSY, HSQC and HMBC) were acquired using 1K data points and 256 increments. 

UV-Vis spectra were recorded with a VersaWave microvolume UV/Vis spectrophotometer (Expedeon, Ottawa, ON, Canada). The molar absorptivity of DOX 10,410 (mol^−1^ L cm^−1^) at 482 nm was used. 

#### 3.2.1. Dynamic Light Scattering and Zeta Potential Measurements

Dynamic light scattering (DLS) and zeta potential (ZP) measurements were performed at 25 °C with a Zetasizer Nano ZS (Malvern Instruments, Oxford, UK) operating at 633 nm (He–Ne laser). The mean hydrodynamic diameter (d) of the NPs was calculated from intensity measurement after averaging the five measurements. The samples (1 mg/mL) were diluted in phosphate buffer (pH = 7.4) prepared in ultrapure water filtered (0.2 µm). 

#### 3.2.2. Mass Spectrometry

MALDI-TOF MS experiments were performed using an AB SCIEX MALDI-TOF/TOF 5800 Analyzer (AB SCIEX, Foster City, CA, USA) equipped with a nitrogen UV laser (λ = 337 nm) pulsed at a 20 Hz frequency by using a set up previously described [26]. Briefly, the mass spectrometer operated in the linear mode and the laser intensity set above the ionisation threshold (4500 in arbitrary units). Mass spectra were processed using Data Explorer 4.11 software (Applied Biosystems, Warrington, UK). 2,5-di-hydroxybenzoic acid (DHB) was used as the matrix, dissolved in water/acetonitrile 1:1 containing 0.03% of CF_3_COOH. Molar-mass averages (M_n_ and M_w_) values were also calculated using Data Explorer software (Applied Biosystems, Warrington, UK).

### 3.3. Cell Culture and Antiproliferative Assay 

A2780 (ovarian carcinoma), A549 (lung carcinoma) and MDA-MB-231 (breast carcinoma) cells (all obtained from ICLC, Genova, Italy) were grown as monolayers in Roswell Park Memorial Institute (RPMI 1640) or Dulbecco’s Modified Eagle’s Medium (DMEM) media (EuroClone, Pero, Italy) supplemented with 10% fetal bovinum serum (FBS) (Euroclone), antibiotics (EuroClone), and non-essential amino-acids (only DMEM, EuroClone). For the assay, cells plated into flat-bottomed 96-well microtiter plates were treated after 6–8 h with the complexes (five 1:5 scalar solutions, 20 µL, starting from 1 µM concentration). Seventy-two hours later, cells were analysed by the 3-(4,5-dimethylthiazol-2-yl)-2,5-diphenyltetrazolium Bromide (MTT) assay as described elsewhere [33]. 

IC_50_ values were calculated from the analysis of single concentration–response curves. Final values are the mean of 4–12 experiments.

### 3.4. Aβ Aggregation Assay

The antiaggregant effect of the newly synthesised compounds on the self-induced aggregation process of Aβ was assayed as previously reported [32]. Briefly, Aβ (20 µM), ThT (60 µM) and the compounds of interest were incubated in phosphate buffered saline (pH 7.4) at 37 °C in a multiplate reader (Varioskan Flash, Thermo Scientific, Leiden, The Netherlands). The fluorimetric readings (excitation and emission wavelengths were 450 nm and 480 nm, respectively) were collected every 10 min up to 60 h. Data of all the measurements, carried out in triplicate, were fitted to Equation (1).
(1)$$F\left(t\right)=F_{0}+\frac{F_{max}-F_{0}}{1+e^{\frac{t-t_{½}}{k}}}$$

*F_max_* − *F*_0_ is the higher fluorescence increment recorded all over the aggregation process; the lag phase (*t_lag_*) is the time interval preceding the formation of amyloid-type species sensitive to ThT. The *t_lag_* values were calculated by using Equation (2)
(2)$$t_{lag}=t_{½}-2/k$$

The parameters of each set of measurements were expressed as the mean ± SD.

### 3.5. Solubility Experiments 

DOX hydrochloride (0.017 M, water solution) was added to the solutions containing different concentrations (25 mg/mL, 12 mg/ML, 6 mg/mL) of all the polymers in phosphate buffer (50 mM, pH 7.4). The suspensions were sonicated for 3 min, incubated at 25 °C in the dark and centrifugated after 2 h. DOX was determined in the supernatant with UV/Vis spectroscopy, at 482 nm. 

### 3.6. Statistical Analysis

For statistical analysis the one way ANOVA was used followed by the post-hoc Bonferroni/Dunn analysis of data.

## Figures and Tables

**Figure 1 molecules-26-01724-f001:**
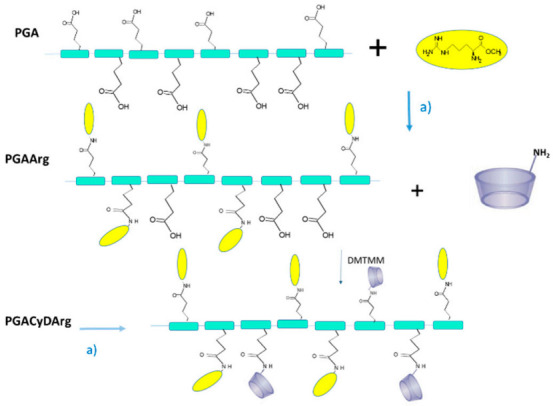
PGA-cyclodextrin (CyD)-Arg polymers, PGA is N-butyl-polyglutamate, a) 4-(4,6-Dimethoxy-1,3,5-triazin 2-yl)-4-methylmorpholinium chloride (DMTMM), H_2_O pH 8, rt, 12 h.

**Figure 2 molecules-26-01724-f002:**
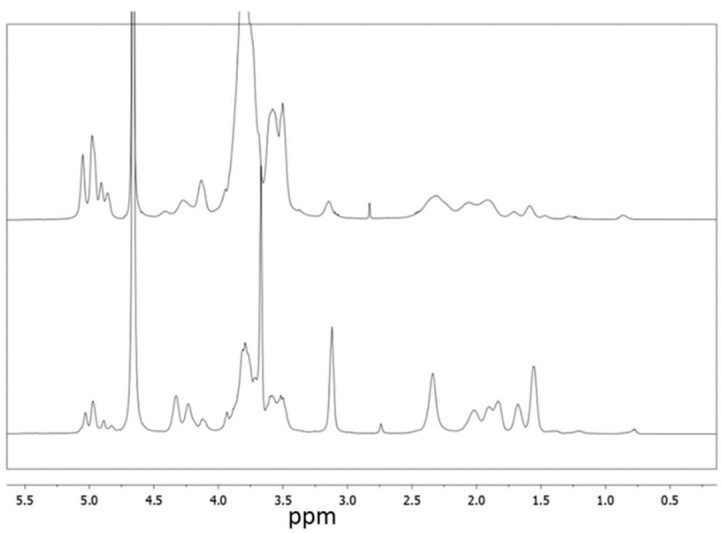
^1^H NMR spectra of PGAβ-CyDArg 1 (top), PGAβCyDArg4 (bottom) (D_2_O, 500 MHz).

**Figure 3 molecules-26-01724-f003:**
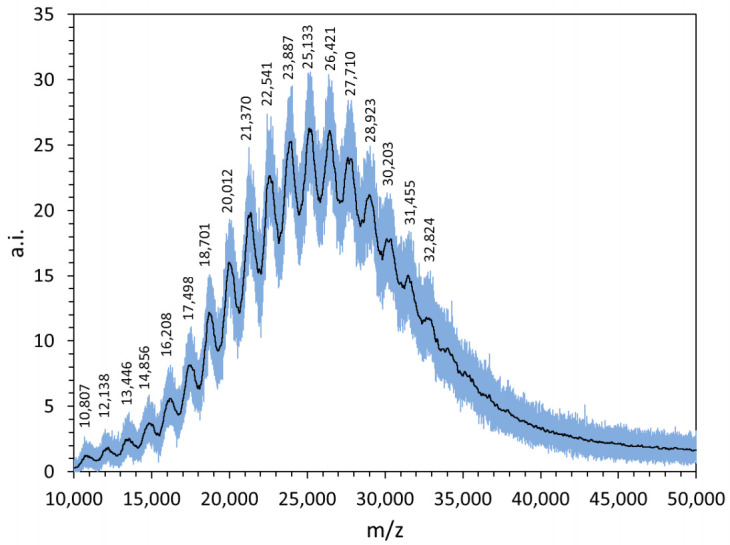
MALDI-TOF MS spectrum of PGAβCyDArg1. The raw spectrum (gray line) was properly smoothed (black line) in order to obtain the *m*/*z* values of all the relative peaks.

**Figure 4 molecules-26-01724-f004:**
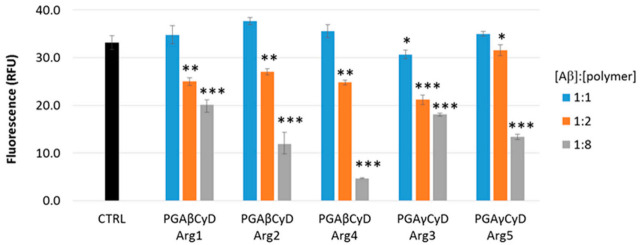
Maximum fluorescence gain values (*F_max_* − *F*_0_) of the samples containing Aβ_1-40_ (20 µM) alone (control (CTRL)) or in the presence of the PGA polymers, the (Aβ)/(Polymer) molar ratio ranging from 1:1 to 1:8. (* *p* < 0.05, ** *p* < 0.01, *** *p* < 0.001 vs. CTRL, ANOVA test).

**Figure 5 molecules-26-01724-f005:**
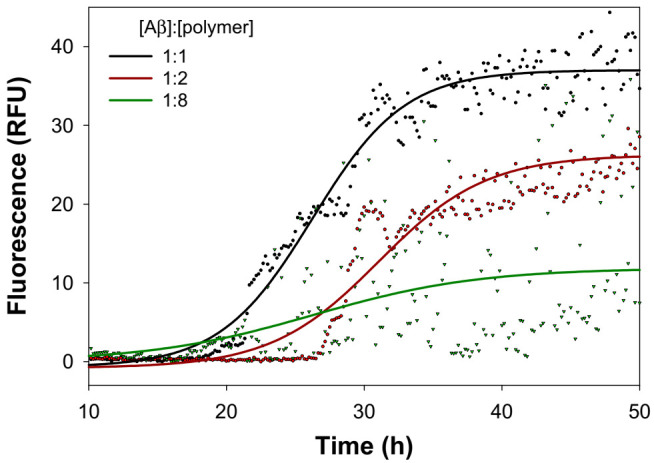
Representative kinetic profiles of the amyloid aggregation due to the co-incubation of Aβ_1-40_ (20 µM) with PGAβCyDArg2, the (Aβ)/(Polymer) molar ratio ranging from 1:1 to 1:8. Single points represent the experimental data, whereas the straight lines are the fitted curves (adjusted R^2^ is 0.9766, 0.9481 and 0.7227 for 1:1, 1:2, 1:8 (Aβ)/(Polymer) molar ratio, respectively).

**Table 1 molecules-26-01724-t001:** Features of PGA-CyD-Arg derivatives.

Polymer	CyD Units	Arg Units	Z Potential (mV)	Mw (Da)
PGAβCyDArg1	19 ± 1	4 ± 1	8 ± 1	25,500
PGAβCyDArg2	15 ± 1	7 ± 1	7.7 ± 0.5	21,700
PGAγCyDArg3	12 ± 1	10 ± 1	2.3 ± 0.5	21,100
PGAβCyDArg4	6 ± 1	15 ± 1	45 ± 5	13,600
PGAγCyDArg5	5 ± 1	15 ± 1	37 ± 3	13,300

**Table 2 molecules-26-01724-t002:** Half maximal inhibitory concentration (IC_50_) values (nM) of doxorubicin (DOX) in the presence of CyD polymers in human tumor cells.

Cell Line	PGAβCyDArg1	PGAβCyDArg2	PGAγCyDArg3	PGAβCyDArg4	PGAγCyDArg5	DOX
A2780 ^a^	4.7 ± 1.7 ^b^	5.9 ± 1.6	12.7 ± 2.4 ^c^	10.0 ± 1.7 ^d^	11.7 ± 0.4 ^e^	7.7 ± 3.9
A549	55.2 ± 10.0	52.2 ± 10.1	70.0 ± 16.6	52.6 ± 4.4	56.2 ± 2.9	54.6 ± 19.2
MDA-MB-231	37.7 ± 11.8	40.9 ± 6.5	50.6 ± 22.5	60.2 ± 15.2	65.3 ± 16.7	40.9 ± 13.8

^a^*p* = 0.0003, as detected by ANOVA; ^b^
*p* = 0.0657 vs. DOX; ^c^
*p* = 0.0001 vs. PGAβCyDArg1; ^d^
*p* = 0.0016 vs. PGAβCyDArg1; ^e^
*p* = 0.0007 vs. PGAβCyDArg1, all calculated by Bonferroni/Dunn post-hoc analysis of data.

## Data Availability

Please refer to suggested Data Availability Statements in the section “MDPI Research Data Policies” at https://www.mdpi.com/ethics.

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
