# Peer review of "Exploring Charged Polymeric Cyclodextrins for Biomedical Applications"

_molecules, 2021, doi:10.3390/molecules26061724_

Round 1

Reviewer 1 Report

The present study reported the synthesis of charged linear cyclodextrin polymers and its biomedical application. The authors characterized the synthetized polymers and study the interaction of cyclodextrin polymers with amyloid beta, demonstrating the prevention of amyloid beta aggregation in function of polymer composition. In addition, the authors tested the synthetized cyclodextrin polymers as drug carriers of anti-cancer therapeutics (doxorubicin) in 2D monolayers of multiple cancer cell lines. The MS was clearly written providing up to date literature references. However, it is anticipated that the authors provide more detailed discussion of the results and the perspectives of cyclodextrin polymers as drug carriers in the comparison with the literature data.

Major points:

The paragraph about the antiproliferative activity requires additional data.

  • Please provide the evidences of the interaction of DOX with cyclodextrin polymers and the comparison of the affinity between the various polymers.

According to this data, the authors could perform the choice the optimal ratio between polymer and DOX

  • Did the authors tested the polymers pre-incubated with DOX for cell toxicity measurements?

  • The description of the results is confusing. The authors reported that “the polymers did not reduce the cytotoxicity of DOX” and “PGAβCyDArg1 … shows a trend towards a higher antiproliferative activity”. However, the decrease of IC50 value compared to free DOX was observed only in A2780 cells (4.7 nM vs 7.7 nM), while for A549 and MDA-MB-231 cells the IC50 values were similar.

At the same time, PGAβCyDArg3 demonstrated dramatic increase of IC50 value (lower cell toxicity) in A2780 and A549 cells (70 nM vs 55nM for free DOX), and PGAβCyDArg4 – for MDA-MB-231 cells (60 nM vs 41 nM). It means that the dose of DOX should be increased in almost 150% to achieve the same cell toxicity. Thus, it may be considered as a reduction of doxorubicin cytotoxicity.

Please clarify the data and the conclusions.

Minor point:

  • Please add the description of the abbreviation (Ab).
  • Please provide the polydispersity index (PDI) for all the DLS samples.
  • Figure 4. Please provide the results of statistical analysis (using ANOVA) for the differences among the various ratios.
  • Figure 5. Please provide the values of fitting errors (e.g. Adjusted R-squared)

Reviewer 2 Report

In the manuscript of N. Bognanni et al. "Exploring charged polymeric cyclodextrins for biomedical applications", the syntheses of new linear polymers of beta-CyD and gamma-CyD with different contents of guanidinium positive charge and number of CyD cavities are reported (five samples). These samples of grafted polymers are characterized by 1H and 13C NMR, MALDI MS profiling, DLS and zeta-potential values. It was shown that these polymers inhibit b-amyloid aggregation. They can be used as drug carriers (doxorubicin was taken for this study; cytotoxicity of DOX was not reduced by PGA-beta-CyD-Arg(1-5)).

Reviewer's notes.

  1. It would be desirable to add a list of abbreviations.
  2. Line 85. Should be: "...or n-butyl chain..." (hyphen is missed.)
  3. Lines 102—103. "The MALDI (mass) spectra recorded in linear mode (Figure S11) mainly contain a single wide band..." This is not so evident: profiles of polymers PGA-beta-CyD-Arg4 and PGA-gamma-CyD-Arg5 have a tendency to bimodality (poorly resolved peaks centered at ca. 15000 and ca. 25000, two upper curves in Fig. S11). How it can be explained?
  4. Line 252: "... referred to D2O signal." Should be: "HDO signal". (Deuterium resonance band is routinely used for field stabilization). What signal was used as an internal standard in 13C NMR spectra? Line 254: HMBC 2D NMR spectra are not reported in Supplementary Materials.
